# Intravenous indocyanine green dye is insufficient for robust immune cell labelling in the human retina

Oliver H. Bell[1], Ester Carreño[2], Emily L. Williams[1], Jiahui Wu[1], David A. Copland[1], Monalisa Bora[2], Lina Kobayter[2], Marcus Fruttiger[3,4], Dawn A. Sim[4], Richard W. J. Lee[1,2,3,4], Andrew D. Dick[1,2,3,4], Colin J. Chu[1,2]*

1 Academic Unit of Ophthalmology, Department of Translational Health Sciences, Faculty of Health Sciences, University of Bristol, Bristol, United Kingdom, 2 Bristol Eye Hospital, University Hospitals Bristol NHS Foundation Trust, Bristol, United Kingdom, 3 UCL Institute of Ophthalmology, London, United Kingdom, 4 Moorfields Eye Hospital NHS Foundation Trust, London, United Kingdom

* colin.chu@bristol.ac.uk

**Data Availability Statement:** All relevant data are within the manuscript, its Supporting Information files, and in repositories. The flow cytometric data

## Abstract

It is not currently possible to reliably visualise and track immune cells in the human central nervous system or eye. Previous work demonstrated that indocyanine green (ICG) dye could label immune cells and be imaged after a delay during disease in the mouse retina. We report a pilot study investigating if ICG can similarly label immune cells within the human retina. Twelve adult participants receiving ICG angiography as part of routine standard of care were recruited. Baseline retinal images were obtained prior to ICG administration then repeated over a period ranging from 2 hours to 9 days. Matched peripheral blood samples were obtained to examine systemic immune cell labelling and activation from ICG by flow cytometry with human macrophage cultures as positive controls. Differences between the delayed near infrared ICG imaging and 488 nm autofluorescence was observed across pathologies, likely arising from the retinal pigment epithelium (RPE). Only one subject demonstrated ICG signal on peripheral blood myeloid cells and only three distinct cell-sized signals appeared over time within the retina of three participants. No significant increase in immune cell activation markers were detected after ICG administration. ICG accumulated in the endosomes of macrophage cultures and was detectable above a minimum concentration, suggesting cell labelling is possible. ICG can label RPE and may be used as an additional biomarker for RPE health across a range of retinal disorders. Standard clinical doses of intravenous ICG do not lead to robust immune cell labelling in human blood or retina and further optimisation in dose and route are required.

## Introduction

It is not currently possible to reliably visualise and track immune cells in human central nervous system tissues. The eye is unique as the only fully-accessible and optically-transparent neural organ in the human body. Indocyanine green (ICG) is a safe and widely-used

is stored in the Flow Repository (http://flowrepository.org/) under the ID: FR-FCM-Z2EV.

**Funding:** This work was supported by a grant from the David Telling Charitable Trust, Bristol. CJC is supported as an NIHR Academic Clinical Lecturer. OHB is funded by the National Eye Research Centre. ELW, DAC, RWJL, ADD, and CJC are supported by the National Institute for Health Research (NIHR) Biomedical Research Centre based at Moorfields Eye Hospital NHS Foundation Trust and UCL Institute of Ophthalmology. The views expressed are those of the Author(s) and not necessarily those of the NHS, the NIHR, or the Department of Health. The funders had no role in study design, data collection and analysis, decision to publish, or preparation of the manuscript.

**Competing interests:** I have read the journal's policy and the authors of the manuscript have the following competing interests: M. Fruttiger and D. A. Sim: Patent (expired). The remaining authors have declared that no competing interests exist.

angiographic dye employed in routine ophthalmic practice for the diagnosis and monitoring of retinal disease [1]. Previously published work identified that ICG administered as a systemic depot injection in the mouse could label myeloid immune cells in the inflamed retina after 48 hours which could be imaged using commercially-available platforms [2]. This approach could advance our understanding of ocular disease and responses to treatment if similar findings can be demonstrated in man.

Equally, it has been shown in patients with rheumatoid arthritis and psoriatic arthritis that joints with active arthritis selectively accumulate more ICG than joints from healthy patients, when measured in the hands of patients and controls [3]. Furthermore, ICG colocalised with macrophages in the lipid-rich atheromas within 20 minutes of intravenous injection in an experimental model in rabbits [4]. This suggests that ICG could selectively accumulate within macrophages and granulocytes.

Following intravenous ICG administration, the main peak of absorption is at a wavelength of 805 nm and the peak of fluorescence emission ranges from 820 to 835 nm, within the near infrared spectrum [5]. It is however well-recognised that the ICG absorption spectra and to a lesser extent the emission spectra, can shift depending on concentration and the other constituents in solution [6]. ICG is injected as an intravenous bolus for ocular angiography up to a maximum of 25 mg in usual practice and retinal images are typically only captured over the first 30-minutes. This means that labelling of cells may not have been observed previously in the human eye, as it is unlikely that cells will have had enough time to be labelled with ICG or to infiltrate the eye if stained peripherally.

ICG is known to enhance the visibility of the retinal pigment epithelium (RPE) mosaic and has been used in adaptive optics imaging [7, 8]. This staining has also been seen late after administration in the normal rat retina [9] in an approach termed delayed near-infra red analysis (DNIRA). It has been suggested this may provide a marker of RPE status in addition to 488 nm autofluorescence but has not been extensively investigated in a clinical setting. With diffuse background staining of RPE by ICG, any signal from immune cell ICG labelling would need to exceed the RPE signal intensity to be detected. This was achieved during retinal inflammation after depot ICG administration in the mouse [2].

Here we report a pilot human study that begins the translation of findings from rodent systems starting by determining if ICG could permit the imaging of immune cells within the human retina using standard intravenous ocular angiographic doses. By extending the imaging period and analysing matched peripheral blood samples we aimed to determine if the currently employed doses and route are sufficient for wider application.

## Materials and methods

### Ethics

The study was prospectively registered with the ISRCTN registry under study number 30128134 (https://doi.org/10.1186/ISRCTN30128134). Ethical approval was obtained from the United Kingdom Health Research Authority and local NHS Research Ethics Committee (17/SW/0030). Human blood samples were handled in accordance with the Human Tissue Act 2004 and the research followed the tenets of the Declaration of Helsinki. Written informed consent was obtained from all participants.

### Recruitment criteria

Twelve adult participants (≥18 years) receiving ICG angiography as part of routine standard of care were recruited. Participants required a suspected diagnosis of neovascular age-related macular degeneration (nAMD), central serous retinopathy (CSR) or intermediate, posterior

and panuveitis. The rationale for combining these diverse patient groups was based on the hypothesis that myeloid immune cells will be present and intraocular leakage of fluid might accumulate ICG at a sufficient concentration within the retina itself.

Ocular inclusion criteria for nAMD or CSR were: 1) likely or suspected choroidal neovascular membrane or central serous retinopathy that clinically required ICG angiography (with or without combined fluorescein angiography); 2) macular sub-retinal fluid present and at least 500 μm in diameter on spectral-domain (SD) optical coherence tomography (OCT) scan; 3) any obscuring haemorrhage should not exceed more than 50% of the area of the sub-retinal fluid; 4) not due for intravitreal injection or photodynamic therapy within the first 48 hours of the study period.

Ocular inclusion criteria for uveitis included: 1) likely or suspected intermediate or posterior uveitis or panuveitis that clinically requires ICG angiography (with or without combined fluorescein angiography); 2) clinically-suspected vasculitis; 3) no intravitreal therapy administered within 3 months prior or will be administered during the study period; 4) mild vitritis only (with a SUN haze score ≤2) to not obscure retinal imaging [10].

The following general exclusion criteria applied: 1) known fluorescein, ICG, iodine, or shellfish allergy; 2) any known contraindication to topical tropicamide and phenylephrine dilating drops; 3) known renal (eGFR ≤80 mL/min/1.73 m$^2$) or hepatic dysfunction or active diseases that in the opinion of the investigator will contraindicate the administration of ICG; 4) inability to be easily imaged on the Spectralis, Optos or Topcon retinal imaging machines (e.g. marked kyphosis or physical impairment); 5) significant media opacity leading to poor image quality (e.g. vitreous haemorrhage or cataract); 6) inability to donate a peripheral blood sample or known HIV, Hepatitis B/C; 7) pregnant or lactating women; 8) unable to consent to study participation or 9) already enrolled in another research trial.

As a pilot study, an exact sample size was not calculated in advance. Participant blood samples were anonymised, and laboratory staff and imaging technicians were masked to clinical details. Image analysis was performed by CJC and EC with full clinical details available.

## Study visits and interventions

Following consent, participants underwent collection of clinical and medication history, followed by ocular and systemic examination with recording of weight, blood pressure, pulse, and temperature. Retinal imaging by colour photography, SD-OCT, infrared reflectance, and 488 nm and 780 nm (standard) and 532 nm and 802 nm fundus autofluorescence (ultra-widefield) was performed at baseline, prior to ICG and fluorescein angiography (where indicated as part of standard of care). The same imaging set was performed after 2, 4, 6, and 8 hours post-ICG injection. Follow-up imaging was performed at a minimum of a further two timepoints of either 24 hours, 48 hours, 7 days, and 9 days post-ICG injection (S1 Table). Each subject received a single injection of ICG at baseline only. No adverse events occurred during the study.

A peripheral blood sample of up to 10 mL was taken from the contralateral arm of that injected with ICG at either 2 or 4 hours post-administration and at a later, broad range of timepoints (24 or 48 hours, or 7 days post-administration) to maximise the chance of detecting a positive signal, given the expected time to labelling was unknown. Where possible following a study amendment, a pre-injection blood sample from the same participant was also obtained for the first flow cytometric run for direct comparison pre- and post-administration–for more robust ICG gating but to also assess potential changes in activation markers of the immune cells to a patient's own baseline. For the second run (days later) fresh healthy donor blood was used as a control, rather than storage and reuse of the participant's own blood, due to potential

depletion of subsets of immune cells as ICG-free blood was required to enable preparation of a fluorescence-minus-one (FMO) control to assists with gating. Blood samples were processed within 30 minutes of venepuncture for flow cytometry.

## Retinal imaging platforms

Retinal angiography and imaging were performed using two different devices: a Spectralis HRA+OCT system (Heidelberg Engineering, Heidelberg, Germany) and an Optos California (Optomap, Optos plc, Scotland, UK). Colour fundus photography used a TRC-50DX camera (Topcon Medical, Tokyo, Japan). SD-OCT imaging was obtained on the Spectralis system.

## Cell preparation

**Human blood-derived macrophages: (hBDMs).** One mM of EDTA and 50 μL/mL RosetteSep human monocyte enrichment cocktail (STEMCELL Technologies, Vancouver, Canada) were added to blood and incubated for 20 minutes at room temperature before dilution with an equal volume of PBS enriched with 2% FCS. Cells were isolated by Ficoll gradient and resuspended at $0.5*10^6$ cells/mL in RPMI complete media (RPMI supplemented with 10% FCS, 2 mM L-glutamine, and 100 U/mL penicillin streptomycin). Fifty ng/mL M-CSF was added and cells were seeded at 2 million CD14$^+$ cells per well of a 6-well plate. Media exchange was performed on days 4, 7 and 10.

**PBMCs.** Whole blood was prepared for ficoll as described above, with the exception that the RosetteSep human monocyte enrichment cocktail was not used. After centrifugation, cells were re-suspended to $0.5*10^6$ cells/mL in freezing media (15% FCS and 10% v/v DMSO in RPMI complete media) and frozen at -80˚C. Cells were thawed by adding pre-heated RPMI complete media prior to use.

**ICG (*in vitro*).** ICG (25 mg powder for solution for injection; PL 44791/0001, Diagnostic Green, Aschheim-Dornach, Germany) was dissolved to a stock solution of 5 mg/mL in pure water before dilution into cell culture media as required.

## Flow cytometry

**hBDMs and whole blood.** One-hundred μL of samples were added to each well of a V-well bottomed plate, 100 μL LIVE/DEAD Fixable Violet Dead Cell Stain (L34955; Thermo Fisher Scientific) was added (to a final dilution of 1:1000 in PBS), and incubated at 4˚C for 15 minutes. Each well was re-suspended with 50 μL of an antibody cocktail and incubated at 4˚C for 30 minutes; two antibody cocktails were used: 1) FITC-conjugated anti-human CD3 mAb (Clone UCHT1; BD Biosciences), PE-conjugated anti-human CD4 mAb (Clone RPA-T4; BD Biosciences), PE-dazzle594-conjugated anti-human CD66b mAb (Clone G10F5; Biolegend, San Diego, CA), PerCP-Cy5.5-conjugated anti-human CD62L mAb (Clone DREG-56; Biolegend), BV605-conjugated anti-human CD25 mAb (Clone BC96; Biolegend), APC-conjugated anti-human CD69 mAb (Clone FN50; Biolegend); 2) PE-dazzle594-conjugated anti-human CD66b mAb, eVolve 605-conjugated anti-human CD14 mAb (Clone 61D3; eBioscience (Thermo Fisher Scientific)), BV711-conjugated anti-human HLA-DR mAb (Clone L243; Biolegend), APC-conjugated anti-human CD80 mAb (Clone 2D10; Biolegend) (all antibodies were diluted in FWB (2% v/v FCS and 1 mM EDTA in PBS); residual volume of blood was considered to be 20 μL). Two-hundred μL of BD FACS lysing solution (349202) was then added and incubated at room temperature for 10 minutes, washed and prepared for flow cytometry acquisition.

For antibody cocktail 1, cells were gated-for on size and granularity (FSC-A vs. SSC-A) and singlets (FSC-A vs. FSC-H), then live cells (LIVE/DEAD Fixable Violet; gate-drawing was

assisted by heat-killed cells), then for CD3$^+$CD4$^+$ (CD4$^+$ T cells), CD3$^+$CD4$^-$ (CD4$^-$ T cells), and CD3$^-$ cells, and (via the CD3$^-$ gate) for CD66b$^+$ (granulocytes), and CD4$^{int}$ cells (monocytes). Cell populations were subsequently gated-for on activation markers using FMO controls (CD25 [11], CD62L [12–14], and CD69 [11]; S1 Fig).

For antibody cocktail 2, live cell singlets were gated-for as described above, and then CD14$^+$ CD66b$^-$ cells (monocytes) were gated-for (CD14 gate-drawing was assisted by FMO controls). The monocytes were subsequently gated-for on activation markers using FMO controls (HLA-DR [15] and CD80 [16]).

For all flow cytometry experiments, 1 μL of mAbs were also added to one drop of compensation beads (Anti-Rat Ig, κ/Negative Control Compensation Particles Set (BD Biosciences)) to prepare single-stain controls for compensation; for live-dead compensation, either heat-killed cells or the ArC Amine Reactive Compensation Bead Kit (Invitrogen) were used. Samples were acquired using a BD LSRFortessa X-20 using BD FACSDIVA software (BD Biosciences).

### Light microscopy

Cells were washed 5 times (for 5 minutes each) in PBS. Images were acquired on an EVOS FL Color Imaging System (Thermo Fisher Scientific).

### Data analysis

Flow cytometry data was analysed and presented using FlowJo v10 (FlowJo LLC, Ashland, OR). Statistical analysis was performed using Prism 7 (GraphPad Software Inc., San Diego, CA). Participant data (percentage positive cells or MFI) was analysed using the Kruskal-Wallis test (one-way ANOVA [17]), with FDR step-up correction of p-values [18].

## Results

It has been identified that systemic ICG administration in the mouse could label myeloid immune cells which were then visualised in the retina [2]. To determine if it could be used for immune cell labelling in man an experimental pilot study was initiated. As ICG is already routinely used for ocular angiography at accepted doses, twelve adults with an existing ocular indication for ICG administration (as part of their standard of care) were enrolled and imaged for up to nine days following angiography. Following a study amendment, from participant 3 onwards the maximum dose of 25mg was always injected. Demographics and the weight-adjusted dose of ICG administered are summarised in Table 1.

### Delayed retinal imaging after ICG administration reveals disparities between standard autofluorescence

Near-infrared (790 nm excitation) retinal autofluorescence signals were minimally detectable or absent at baseline in most of the patients, but clearly visible in all following ICG administration, even nine-days later (S1 Fig). Differences in signal patterns between this DNIRA and standard 488 nm autofluorescence could be seen across a range of ocular pathology (Fig 1) and may provide an additional method for visualising RPE and cell health.

### ICG labelling of Human immune cells *in vivo* is not robust using standard intravenous doses

Only one subject demonstrated ICG signal on peripheral blood immune cells using flow cytometry. Angiography confirmed an acute choroidal neovascular membrane (CNV) which

**Table 1. Demographics of participants enrolled in the study with a clinical indication for routine ICG angiography.**

| ID | Age (years) | Gender (M/F) | ICG indication and eye | Weight (kg) | Dose of ICG* (mg/kg) | Lens Status | Visual Acuity RE (Letter score) | Visual Acuity LE (Letter score) | Past Medical History | Current systemic medication | Recent ocular medication | ICG+ cells detected in blood |
|---|---|---|---|---|---|---|---|---|---|---|---|---|
| 1 | 76 | M | nAMD LE | 96 | 0.08 | C | 73 | 62 | Treated Bladder and Prostate Carcinoma | Atorvastatin | Aflibercept 2 days prior | NO |
| 2 | 78 | F | nAMD BE | 58 | 0.30 | IOL | 72 | 78 | None | Multivitamin supplements | Lucentis 6 months prior, previous PDT | NO |
| 3 | 79 | F | nAMD RE | 112 | 0.22 | C | 63 | 81 | Hysterectomy and stoma from bowel injury. TIA | Lutein | None | YES |
| 4 | 22 | F | Uveitis BE (MFC) | 125 | 0.20 | N | 75 | 2 | None | Tacrolimus, Mycophenolate mofetil | None | NO |
| 5 | 79 | F | nAMD RE | 75 | 0.33 | C | 42 | 79 | None | None | Aflibercept one month prior | NO |
| 6 | 64 | F | nAMD LE | 65 | 0.38 | N | 78 | 34 | Depression, hypertension | Paroxetine | None | NO |
| 7 | 76 | F | Uveitis BE (Panuveitis) | 60 | 0.42 | IOL | 83 | 65 | Ocular sarcoidosis | None | None | NO |
| 8 | 30 | M | Uveitis BE (Panuveitis) | 84 | 0.30 | IOL | 83 | 85 | Arthritis of the knee | None | Ozurdex one month prior & topical prednisolone | NO |
| 9 | 86 | F | nAMD LE | 87 | 0.29 | IOL | 76 | 6 | Hiatus hernia, osteoporosis, hypertension, hypercholesterolaemia | Lansoprazole, Adcal D3, Lisinopril, Cefalexin | Aflibercept six months prior | NO |
| 10 | 78 | M | CSR BE | 72 | 0.35 | N | 37 | 60 | Polymyalgia rheumatica, hypertension, Diabetes Mellitus, Myeloma | Prednisolone, Amlodipine, Atorvastatin, Metformin, Cholecalciferol, Ramipril | None | NO |
| 11 | 34 | F | Uveitis BE (Panuveitis) | 56 | 0.45 | IOL | 79 | 60 | Tuberculosis, Sarcoidosis | Rifater, Moxifloxacin, Pyridoxime | Dexamethasone & Dorzolamide drops | NO |
| 12 | 74 | F | nAMD LE | 90 | 0.28 | N | 78 | 15 | None | Atorvastatin, Clopidogrel, Indapamide MR | None | NO |

ID = identification number, nAMD = neovascular age-related macular degeneration, CSR = Central serous retinopathy, MFC = Multifocal Choroiditis with Panuveitis, LE = Left eye, RE = Right eye, BE = Both eyes, N = No cataract, C = cataract, IOL = intraocular lens, TIA = transient ischaemic attack.

*Note total ICG dose injected was increased to a fixed maximal 25mg from participant 3 onwards.

had been symptomatic for five days prior to ICG injection (Fig 2A–2D). Discrete signals around the CNV lesion were visible only in the near-infrared channel and were approximately 30 μm in diameter compatible with myeloid cells (Fig 2E and 2F). Longitudinal observation demonstrated alteration in position of these signals between the 24 hour and 7 day imaging intervals without corresponding changes in 488 nm autofluorescence, indicating the changes are unlikely to arise from the RPE. It is possible these may represent labelled cells, but the only characteristic possible to confirm this would be movement in position across time, though the

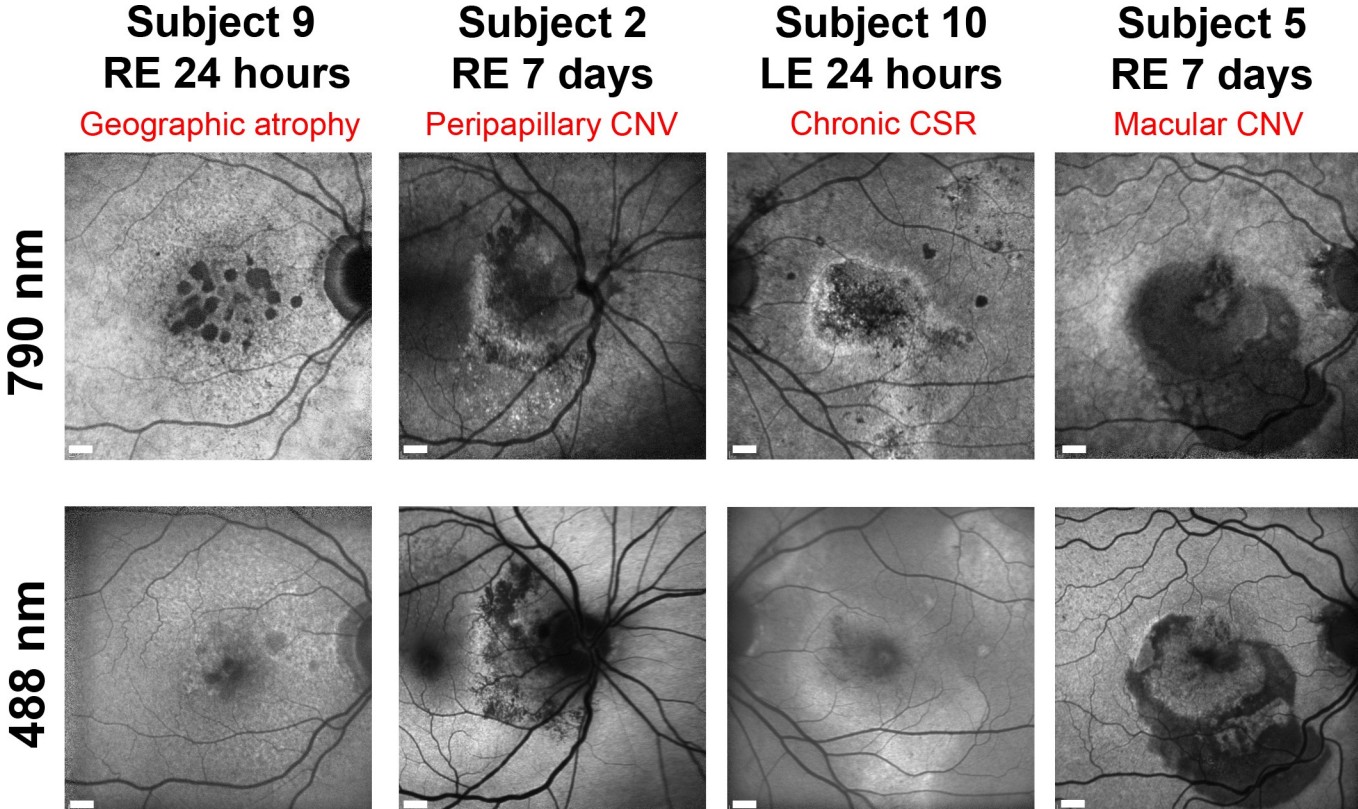

**Subject 9**
**RE 24 hours**
Geographic atrophy

**Subject 2**
**RE 7 days**
Peripapillary CNV

**Subject 10**
**LE 24 hours**
Chronic CSR

**Subject 5**
**RE 7 days**
Macular CNV

790 nm

488 nm

**Fig 1. Differences in retinal images between late near infrared imaging following ICG (DNIRA) and standard 488 nm autofluorescence.** Four examples from the eyes of different subjects illustrate different appearances that provide complementary information on RPE health. Retinal ICG fluorescence 790 nm excitation channel images are shown alongside matched 488 nm excitation autofluorescence images. CNV = Choroidal neovascular membrane, CSR = Central serous retinopathy. LE = left eye, RE = right eye. Scale bars = 600 μm.

speed of change is unknown. There was no evidence of systemic infection or inflammation in this participant to explain why ICG signal was present in the blood.

Compensating for the potential confounding factor of signal arising from RPE dysfunction and the expected rare occurrence of labelled immune cells using the intravenous ICG route, it was possible to identify distinct cell-sized signals (Fig 3) in only three locations of three subjects. In each, the DNIRA signals newly-appear over time in a region where there is no alteration in 488 nm autofluorescence. No ICG signal was observed in the retinal vasculature at late timepoints, in agreement with published data, showing it has a relatively short half-life of minutes in the circulation [19, 20].

### No peripheral blood immune cell activation was observed following ICG administration

For a subset of participant blood samples, it was possible to measure activation markers using flow cytometry (Fig 4, S2 and S3 Figs). Aggregate data did not demonstrate evidence of ICG resulting in an inflammatory or activated immunophenotype. As only one subject had detectable ICG signal in peripheral blood, *in vitro* human macrophage cell cultures were used to examine their ability to develop ICG-positive signals on flow cytometry. Microscopy demonstrated endosomal accumulation of ICG in both cell cultures at an *in vitro* dose of 0.2 mg/ml, implying a sufficient concentration or protracted exposure may be required to achieve labelling where cells would be clearly visible if they gain entry into the retina.

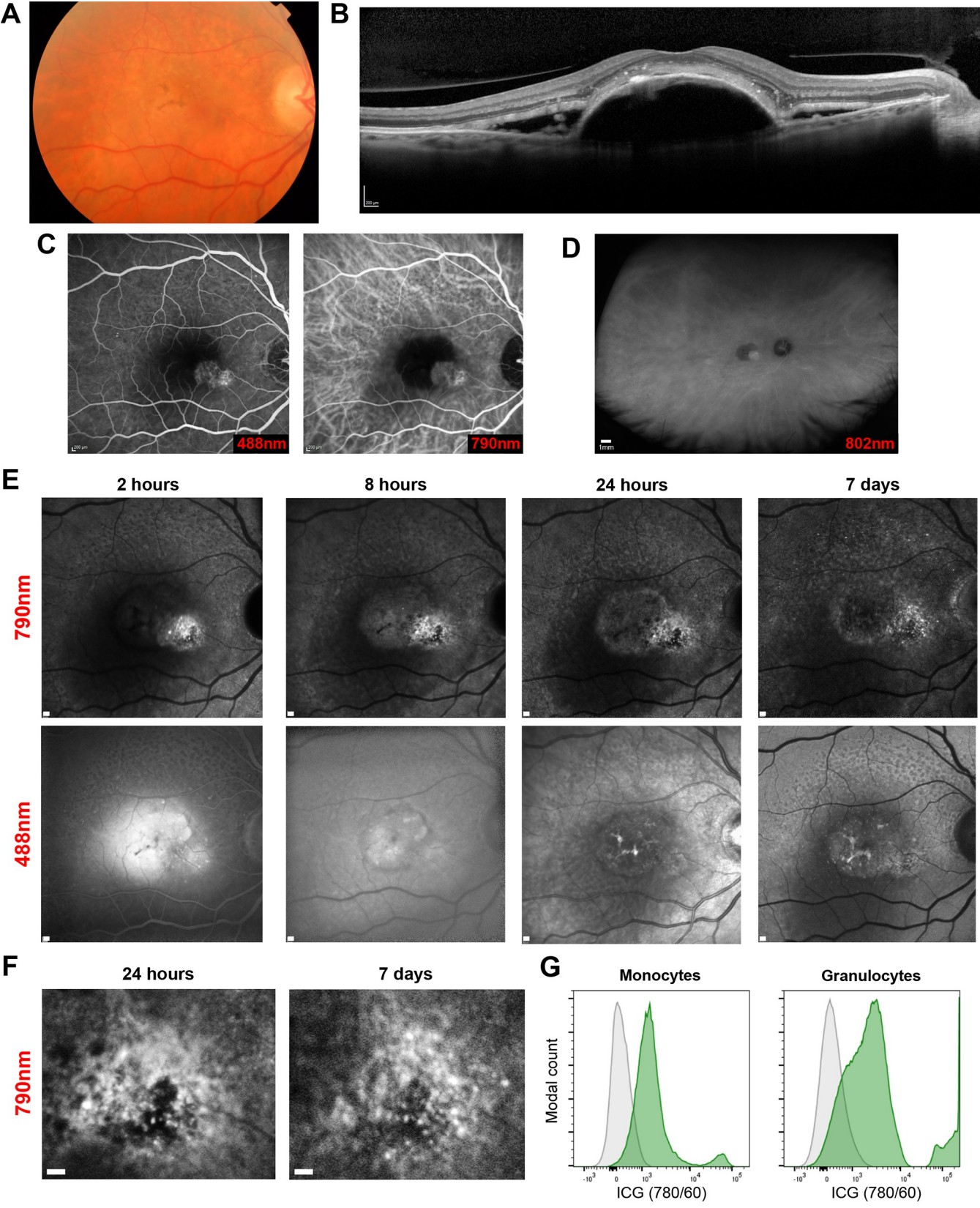

**Fig 2. Only one subject demonstrated ICG-positive cells in peripheral blood and retinal signals that altered over time.** (A) Subject 3 presented a week following new onset visual loss and distortion in the right eye. Existing RPE pigment abnormalities were visible on colour fundus images, (B) with a large pigment epithelial detachment and subretinal fluid visible on spectral-domain OCT. (C) A choroidal neovascular membrane was confirmed by fluorescein and ICG angiography, at 90 and 40 seconds post-injection respectively. (D) Optos ultra-widefield ICG imaging (802 nm excitation) confirmed regions of leakage or high signal were present only at the macula. (E) At different points timed from ICG administration, Spectralis HRA imaging demonstrated punctate signals in the ICG channel only (790 nm excitation) that altered markedly between 24 hours and day 7. (F) Enlarged region illustrating changes in signal intensity and location. (G) Flow cytometry histogram of the peripheral blood sample taken at 24 hours demonstrating ICG+ signal (green) on monocyte and granulocyte populations compared to a no-ICG control (grey). Scale bars = 200 μm unless stated.

## Discussion

Whilst immune cell labelling was the central objective, we illustrate a range of retinal pathology can be imaged late after ICG administration, with distinctive signals arising from the RPE. DNIRA has been described before, but here we demonstrate findings across a further diverse range of human ocular pathology using a commercially available SLO platform [9] that complements recent findings using adaptive optics imaging of the RPE in inherited retinal degenerations [8]. Several examples of unique patterns only seen by DNIRA identify additional or complementary information for monitoring RPE health versus traditional 488 nm autofluorescence imaging. These data highlight an opportunity for superior stratification and earlier detection of pathology such as reversible RPE dysfunction, critical to identifying therapeutic response signals, as has remained problematic for example in studies of geographic atrophy [21].

In terms of immune cell labelling, using the intravenous route with standard angiographic doses does not achieve robust labelling within the retina. ICG-positive myeloid cells were only detected in the peripheral blood of subject 3, in which distinct near infrared punctate signals appeared without corresponding 488 nm signal alterations. These new signals of the appropriate size for a leukocyte changed over time and might represent immune cell trafficking of an ICG-labelled cell. Two similar signals were only seen in two other eyes of two different participants. The only feasible non-invasive method of characterising them as immune cells in man would be to identify clear movement over a period of minutes to hours. Co-labelling with

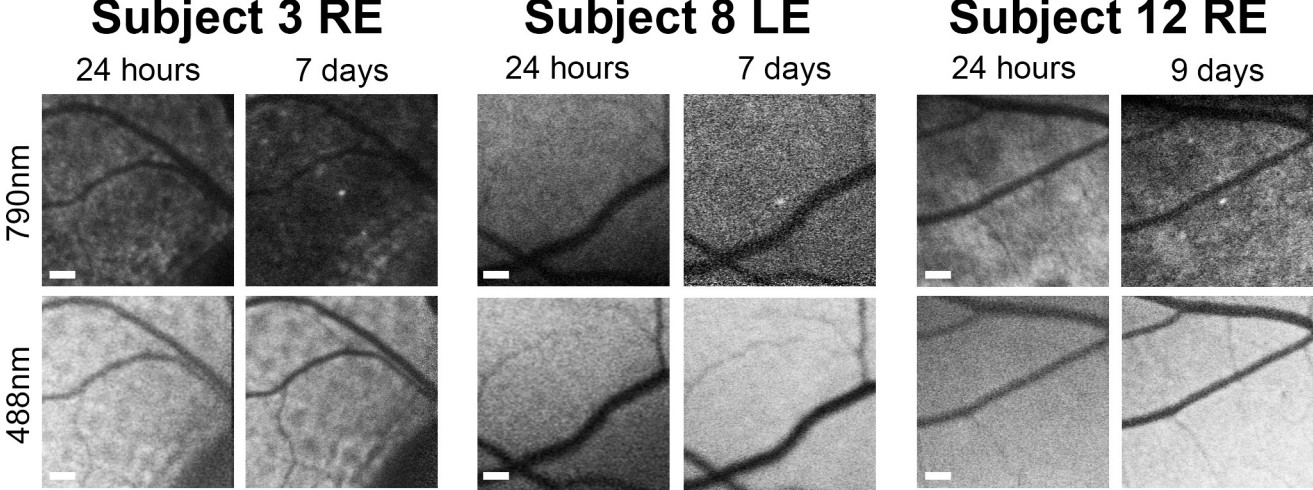

**Fig 3. Cell-sized hyperfluorescent near-infrared signals appearing in regions with no prior autofluorescence or RPE signal change during the study.** Retinal ICG fluorescence 790 nm excitation channel images are shown alongside matched 488 nm excitation autofluorescence images. Subject 3 had detectable ICG cell staining in peripheral blood. By size and location these could represent weakly ICG labelled cells infiltrating the retina, rather than arising from RPE staining alone. LE = left eye, RE = right eye. Scale bars = 200 μm.

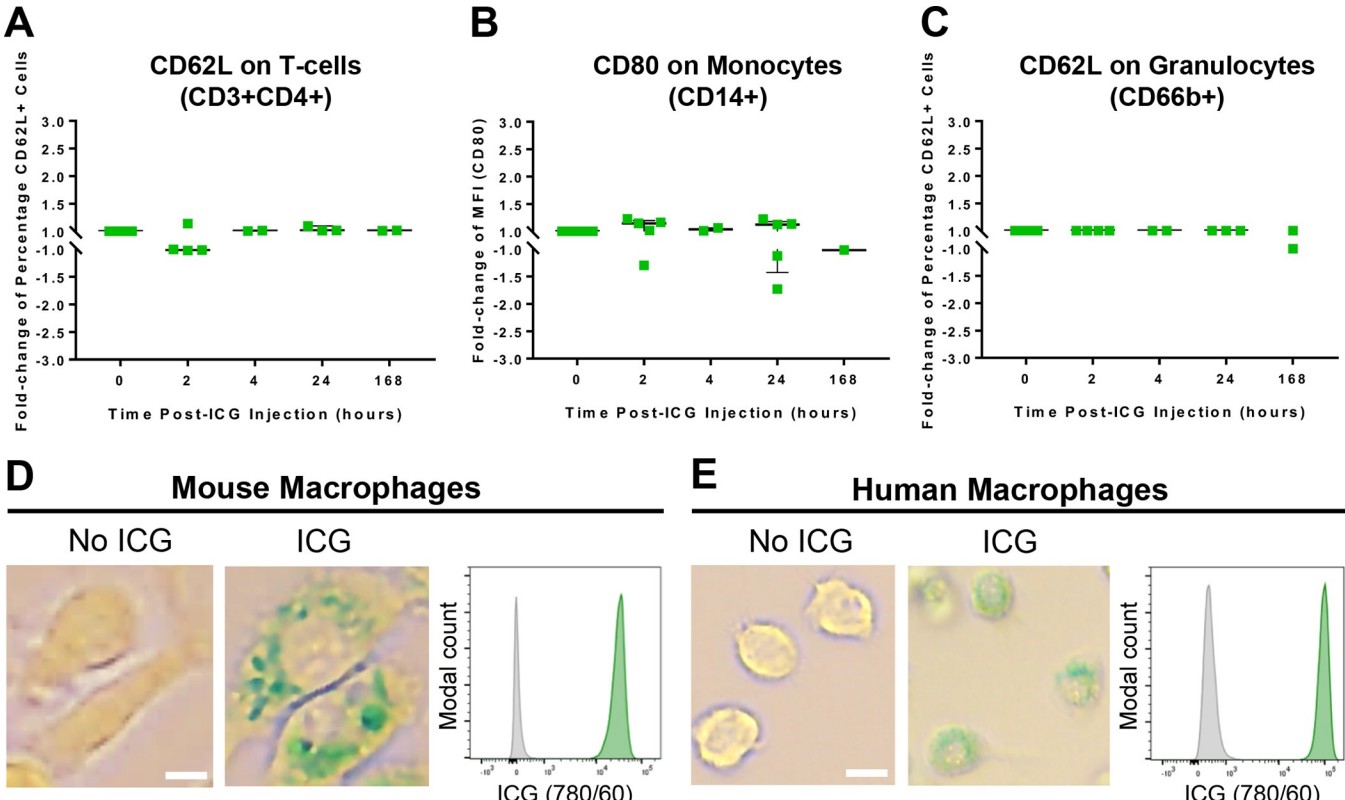

**Fig 4. ICG administration did not activate immune cells in peripheral blood but can be observed in endosomes of macrophage cell cultures.** (A) Aggregated flow cytometry activation marker changes in participant peripheral blood samples taken across the study are shown measuring CD62L on T-cells (p = 0.914), (B) CD80 on monocytes (p = 0.914), and (C) CD62L on granulocytes (p = 0.698). Each point is a single blood sample from 7 subjects. All activation markers underwent validation with positive control stimuli. Human macrophage cell cultures developed visible signal within endosomes following 24 hours of culture with 0.2 mg/mL ICG by microscopy, which could be detected by flow cytometry in the near infrared channel. (D) Representative images shown for human cell cultures, with (green histogram) and without (grey histogram) ICG administration. Scale bars = 10 μm.

cellular dyes or antibodies are not currently established. No rapid alterations were seen during imaging sessions, potential cell labelling could only be identified in post-processing and the expected speed of migration within retinal tissue remains unknown. Evidence from 2-photon imaging of the inflamed mouse eye however demonstrates myeloid cell movement may be much slower than expected, even compared to other tissues, at around 1 μm per minute [22]. This could explain the larger time intervals required to observe signal position shifts that could represent labelled immune cells. Complementing our data and drug safety, no evidence of immune cell activation could be detected in peripheral blood following intravenous ICG injection.

To better understand the pharmacodynamics of ICG in relation to cellular accumulation, we tested human-derived macrophage cell cultures which confirmed the localisation of ICG within extranuclear vesicular structures. Whilst not directly comparable to the *in vivo* situation, this suggests if the correct delivery and concentration can be achieved, labelling is possible. The intracellular localisation is in contrast with other cells such as hepatocytes, where nuclear localisation has been observed, suggesting differential uptake or processing routes [23]. The uptake of ICG has been shown, in primary human RPE cultures, to be dependent on the $Na^+/K^+$-ATPase pump and localised within extranuclear compartments [24]. It has also been observed to bind the RPE of human, primate, and mouse subjects *in vivo* [9, 25]. In

Subject 3, several cell types were weakly-positive for ICG and this is likely due to membrane binding as ICG is well-characterised as able to bind to amphiphilic structures [26]; however the phagocytic populations (monocytes and granulocytes) were far more strongly-positive for ICG than other immune cells. This suggests that ICG may be internalised and accumulated by these cells, comparable to that observed in our macrophage cell cultures and may therefore be the only visible population brighter than the RPE if they enter the retina.

Our findings highlight the potential of ICG, though considerably more refinement is needed prior to further clinical application because we identified ICG-positive blood in only one subject at 24 hours post-administration. It was not possible to test peripheral blood at every imaging visit, so positive findings may have been missed. Preclinical mouse data suggested the intravenous route of ICG administration would not provide the most efficient cell labelling, but it was prudent from an ethical and practical perspective to show the established administration route was insufficient [2]. The results of this study provide justification to pursue alternative methods of administration, reformulation or higher doses.

In clinical hepatic function studies, approximately ten-fold higher systemic ICG doses than used here have been given successfully without clear evidence of toxicity [27]. This dose has typically only been given as a bolus and given the endosomal accumulation and preferential staining of myeloid cells with phagocytic potential, it is likely more sustained exposure of cells to a minimal concentration of ICG would be needed for efficient labelling. The superior intra-peritoneal or subcutaneous routes identified in mouse are not readily applicable in man given the safety, dose, and side-effect considerations for ocular application [2]. However, administration by slow intravenous infusion or reformulation to an oral preparation may be preferable and our *in vitro* modelling suggests a sustained blood level of even 6 μg/ml of ICG for two hours might enable strong cell marking to be reliably detectable using flow cytometry (S2 Fig).

With improvement in the efficiency of cell labelling, the relative amplitude of signal from immune cells against that of the RPE could result in clear separation of the two, as achieved in the mouse [2]. Furthermore, engaging methods of superior imaging resolution such as adaptive optics scanning laser ophthalmoscopy will allow single immune cells to be clearly distinguished from the RPE, identified and tracked. This study provides further support towards the translational goal of imaging the cellular components of the immune system in the human eye and with considerable refinement ICG might be a viable method to achieve this.

## Supporting information

**S1 Table. Imaging session and blood sample collection timings for each participant.** Measured from time of ICG injection. Timings were determined by participant availability and following an amendment, extension to cover a range of longer intervals to maximise the chance of signal detection.
(DOCX)

**S1 Fig. Baseline 488 and 790 nm fundal images.** A representative selection of baseline images acquired from patients in the study illustrating weak to minimal infrared autofluorescence prior to ICG administration. Scale bars = 300 μm.
(TIF)

**S2 Fig. Representative gating strategy used to analyse human blood.** (A) The representative gating strategy used for antibody cocktail #1 ("lymphocyte panel") is shown. For antibody cocktail #2 ("monocyte panel"), live cells were gated into granulocytes (CD66b$^+$) or monocytes (CD14$^+$ CD66b$^-$, a CD14 FMO was used to assist with drawing the CD14$^+$ gate)). For the histograms (activation markers: CD25, CD62L, and CD69), FMOs are indicated in red, whilst a

representative participant sample is indicated in blue. Validation of the activation marker antibodies was performed via whole-blood staining (individually for 24 hours) with phytohaemagglutin P (PHA; at a final concentration of 20 μg/mL), CD3/CD28 activating beads (Dynabeads; 0.63 μL of beads (stock concentration $4.0 * 10^7$ beads/mL) were used), and li popolysaccharide (LPS; at a final concentration of 0.1 ng/mL). (B) Representative histograms from an experiment where whole human blood was stained *ex vivo* with 6 ug/mL indocyanine green (ICG) for 2 hours (green), as compared to an ICG FMO sample (grey). The staining is less bright than has been observed with cell lines and the participant data, possibly due to the abundance of erythrocytes (as compared-to cell cultures), lower ICG concentration, and/or shorter staining time.
(TIF)

**S3 Fig. Scatterplots showing changes in activation after ICG administration.** (A–B) Scatterplots for $CD3^+CD4^+$ cells show changes in CD25 and CD69, (C–E) for $CD3^+CD4^-$ cells show changes in CD25, CD62L, and CD69, and (F) for $CD14^+$ cells (monocytes) show changes in HLA-DR. The y-axis is expressed as fold-change in the percentage of positive cells (CD25, CD62L, CD69) or the fold-change of the MFI (CD80, HLA-DR). Loss of CD62L is associated with activation, whilst expression of CD25 and CD69, and up-regulation of CD80 and HLA-DR associate with activation; statistical tests were not significant (p $\geq$ 0.05). The data was acquired using flow cytometry (n = 7).
(TIF)

## Acknowledgments

The Authors wish to thank Andrew Herman and Lorena Ballesteros (Flow Cytometry Facility, University of Bristol) for technical assistance; Eleanor Hiscott, Julie Cloake and the staff of the Bristol Eye Hospital Clinical Research Unit, and Optos Plc who kindly supplied the Optos California machine for the duration of this study.

## Author Contributions

**Conceptualization:** Ester Carreño, Marcus Fruttiger, Dawn A. Sim, Richard W. J. Lee, Andrew D. Dick, Colin J. Chu.

**Data curation:** Oliver H. Bell, Monalisa Bora.

**Formal analysis:** Oliver H. Bell, Ester Carreño, Colin J. Chu.

**Funding acquisition:** Ester Carreño, Colin J. Chu.

**Investigation:** Oliver H. Bell, Ester Carreño, Emily L. Williams, Jiahui Wu, David A. Copland, Monalisa Bora, Colin J. Chu.

**Methodology:** Emily L. Williams.

**Project administration:** Oliver H. Bell, Ester Carreño, Monalisa Bora, Colin J. Chu.

**Resources:** Emily L. Williams, David A. Copland, Monalisa Bora, Richard W. J. Lee, Andrew D. Dick.

**Supervision:** David A. Copland, Richard W. J. Lee, Andrew D. Dick, Colin J. Chu.

**Validation:** Oliver H. Bell, Emily L. Williams.

**Visualization:** Oliver H. Bell, Ester Carreño, Lina Kobayter, Colin J. Chu.

**Writing – original draft:** Oliver H. Bell, Ester Carreño, Colin J. Chu.

**Writing – review & editing:** Oliver H. Bell, Ester Carreño, Emily L. Williams, Jiahui Wu, David A. Copland, Monalisa Bora, Lina Kobayter, Marcus Fruttiger, Dawn A. Sim, Richard W. J. Lee, Andrew D. Dick, Colin J. Chu.

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
