## [Decision Letter · Decision Letter 0]

7 Jan 2020

PONE-D-19-32243

Intravenous indocyanine green dye is insufficient for robust immune cell labelling in the human retina

PLOS ONE

Dear Mr Bell,

Thank you for submitting your manuscript to PLOS ONE. After careful consideration, we feel that it has merit but does not fully meet PLOS ONE’s publication criteria as it currently stands. Therefore, we invite you to submit a revised version of the manuscript that addresses the points raised during the review process.

Both reveiwers found the work interesting in general. Although one Reviewer has some minor methodological Points to be adressed does the other Reviewer ask for some more data and a focus of these data on one subgroup  of patients; here neovascular AMD. That Revision shoud however, Keep the other disease subtypes in the paper.

We would appreciate receiving your revised manuscript by Feb 21 2020 11:59PM. If you would like to make changes to your financial disclosure, pleaBVse include your updated statement in your cover letter.

To enhance the reproducibility of your results, we recommend that if applicable you deposit your laboratory protocols in protocols.io, where a protocol can be assigned its own identifier (DOI) such that it can be cited independently in the future. For instructions see: http://journals.plos.org/plosone/s/submission-guidelines#loc-laboratory-protocols

We look forward to receiving your revised manuscript.

Kind regards,

Olaf Strauß

Academic Editor

PLOS ONE

Journal Requirements:

2. We note that you have provided the following statement in your manuscript "The treatment of animals conformed to the ARVO Statement for the Use of Animals in Ophthalmic and Vision Research.". Please ensure information provided in the manuscript only pertains the present study.

"I have read the journal's policy and the authors of the manuscript have the following

competing interests: M. Fruttiger and D.A. Sim: Patent (expired).

The remaining authors have declared that no competing interests exist."

Reviewers' comments:

Reviewer's Responses to Questions

**Comments to the Author**

1. Is the manuscript technically sound, and do the data support the conclusions?

Reviewer #1: Partly

Reviewer #2: Yes

2. Has the statistical analysis been performed appropriately and rigorously? 

Reviewer #1: N/A

Reviewer #2: Yes

3. Have the authors made all data underlying the findings in their manuscript fully available?

Reviewer #1: Yes

Reviewer #2: Yes

4. Is the manuscript presented in an intelligible fashion and written in standard English?

Reviewer #1: Yes

Reviewer #2: Yes

5. Review Comments to the Author

Reviewer #1: The manuscript follows up on a previously published paper from the same group that used ICG imaging to detect retinal immune cells in mice.

Despite its mostly negative findings that ICG labeling as it stands now is insufficient for robust immune cell labeling in the human retina, the paper is principally very interesting and important. However, the major problem of the study is the limited number of enrolled patients (n=12) and the mixed disease population (nvAMD, Uveitis, CSR). To come to a robust conclusion at least for one disease entity, the authors should limit themselves to one major disease group (e.g. nvAMD under different treatment regimens) and increase the sample number. It is likely that the number of subjects that shows ICG signals in the retinal will then be increased. Also, a correlation of ICG signals with treatment could be interesting, as it is well known that anti-VEGF /anti-PGF treatment may influence retinal immune cell homeostasis.

Reviewer #2: In the manuscript, entitled:” Intravenous indocyanine green dye is insufficient for robust immune cell labelling in the human retina”, the authors took effort to determine if immune cell infiltration could be detected by routine clinical imagining techniques. Having such an opportunity would help to better understand time course and pathology of ocular inflammation in different eye diseases. This could also be a valuable tool to track treatment related to inflammation as well as to complications related to a leaky blood retina barrier. Unfortunately, only 1 patient demonstrates ICG labeled cells in the blood and in the retina. The authors mention that the low success rate may be due to the low dosage of systemic ICG used for ophthalmic imaging and that increasing the dose might lead to better labelling. The manuscript is interesting, very well written, technically correct and will hopefully pave the way for follow up studies. I have only a few comments, listed below, which should be addressed before I could recommend this manuscript for publication in Plos One.

Line 29: “Matched peripheral blood samples…..” It is not quite clear why those samples were analyzed. Please add a half sentence to explain that this experiment was done to determine if ICG labels immune cells.

Line 209, Result section: Adding headlines might provide more guidance through the section and help improve the structure of the manuscript.

Line 220, Table1: Is there an explanation why patient 1 has only received 0.08 mg/kg ICG? Please also introduce all abbreviations (LE, BE, RE, etc) in the legend with their first occurrence.

Line 270, Figure 3 legend: The authors state that subject 3 shows ICG positive cell labeling in PBMCs and that ICG positive cells can be found in the retinal parenchyma. However, they fail to explain why there is no ICG signal present in the vasculature. Further explanation of this could prove informative.

Line 463, S1 Figure legends: The figure legends don’t line up with the images and parts of the graphs in S2 are missing, at least in my copy.

6. PLOS authors have the option to publish the peer review history of their article (what does this mean?). If published, this will include your full peer review and any attached files.

Reviewer #1: No

Reviewer #2: No

---

## [Author Response · Author response to Decision Letter 0]

22 Jan 2020

We enclose revisions as requested, and thank the Reviewers for their feedback.

---

## [Decision Letter · Decision Letter 1]

30 Jan 2020

Intravenous indocyanine green dye is insufficient for robust immune cell labelling in the human retina

PONE-D-19-32243R1

Dear Dr. Chu,

We are pleased to inform you that your manuscript has been judged scientifically suitable for publication and will be formally accepted for publication once it complies with all outstanding technical requirements.

With kind regards,

Olaf Strauß

Academic Editor

PLOS ONE

Additional Editor Comments (optional):

Reviewers' comments:

Reviewer's Responses to Questions

**Comments to the Author**

1. If the authors have adequately addressed your comments raised in a previous round of review and you feel that this manuscript is now acceptable for publication, you may indicate that here to bypass the “Comments to the Author” section, enter your conflict of interest statement in the “Confidential to Editor” section, and submit your "Accept" recommendation.

Reviewer #1: All comments have been addressed

Reviewer #2: All comments have been addressed

2. Is the manuscript technically sound, and do the data support the conclusions?

Reviewer #1: Yes

Reviewer #2: Yes

3. Has the statistical analysis been performed appropriately and rigorously? 

Reviewer #1: Yes

Reviewer #2: Yes

4. Have the authors made all data underlying the findings in their manuscript fully available?

Reviewer #1: Yes

Reviewer #2: Yes

5. Is the manuscript presented in an intelligible fashion and written in standard English?

Reviewer #1: Yes

Reviewer #2: Yes

6. Review Comments to the Author

Reviewer #1: (No Response)

Reviewer #2: (No Response)

7. PLOS authors have the option to publish the peer review history of their article (what does this mean?). If published, this will include your full peer review and any attached files.

Reviewer #1: Yes: Thomas Langmann

Reviewer #2: No

---

## [Editor Report · Acceptance letter]

7 Feb 2020

PONE-D-19-32243R1 

Intravenous indocyanine green dye is insufficient for robust immune cell labelling in the human retina 

Dear Dr. Chu:

I am pleased to inform you that your manuscript has been deemed suitable for publication in PLOS ONE. Congratulations! Your manuscript is now with our production department. 

With kind regards,

on behalf of

Professor Olaf Strauß 

Academic Editor

PLOS ONE